# Preparation of Bio-Based Polyurethane Coating from *Citrullus colocynthis* Seed Oil: Characterization and Corrosion Performance

**DOI:** 10.3390/polym16020214

**Published:** 2024-01-11

**Authors:** Ahmed S. Alshabebi, Maher M. Alrashed, Lahssen El Blidi, Sajjad Haider

**Affiliations:** Chemical Engineering Department, College of Engineering, King Saud University, P.O. Box 800, Riyadh 11421, Saudi Arabia; 439106452@student.ksu.edu.sa (A.S.A.); mabdulaziz@ksu.edu.sa (M.M.A.); shaider@ksu.edu.sa (S.H.)

**Keywords:** *Citrullus colocynthis*, epoxidation, polyol, polyurethane, surface coating, anticorrosion, renewable sources

## Abstract

In this study, a new epoxidized oil from *Citrullus colocynthis* seed oil (CCSO) was obtained for a potential application in the formulation of polyurethane coatings. Initially, the fatty acid composition of CCSO was determined by gas chromatography–mass spectrometry (GC–MS). Subsequently, the epoxidation of CCSO was performed with in situ generated peracetic acid, which was formed with hydrogen peroxide (30 wt.%) and glacial acetic acid and catalyzed with sulfuric acid. The reaction was continued at a molar ratio of 1.50:1.0 of hydrogen peroxide to double bond (H_2_O_2_:DB) for 6 h at a controlled temperature of 60 °C. The resulting epoxidized oil was then used to produce a bio-based polyol by hydroxylation. The molar ratio of epoxy groups to methanol and distilled water was maintained at 1:11:2, and the reaction was carried out for 2 h at a controlled temperature of 65 °C. The major functional groups of the epoxidized oil and its polyol were validated by Fourier-transform infrared (FT-IR) and proton nuclear magnetic resonance (^1^H NMR) spectroscopies. A polyurethane (PU) coating was produced from the synthesized polyol and 3HDI isocyanurate, keeping the molar ratio of NCO:OH at 1:1. The resulting PU coating was then applied to glass and aluminum panels (Al 1001). After the film was cured, the properties of the PU coating were evaluated using various techniques including pencil hardness, pendulum hardness, adhesion, gloss, chemical resistance, and EIS tests. The results show that the PU coating obtained from CCSO is a promising new raw material for coating applications.

## 1. Introduction

The increasing use of plants, vegetations, and animal-derived materials increases the release of greenhouse gases, contributes to the overfilling of landfills, and depletes non-renewable resources [1]. This situation prompts us to look for new alternatives to mitigate the increasing production of fossil materials. Vegetable oils (VOs) are a particularly promising renewable source due to their wide availability, cost effectiveness, and lack of toxicity [2].

Vegetable oils (VOs) comprise triglycerides, which are compounds formed from glycerol and three fatty acids. The fatty acid chains can vary in length from 8 to 24 carbons and may have 0 to 5 carbon–carbon double bonds [3]. VOs are commonly used to make lubricants, cosmetics, surfactants, paints, coatings, and resins. They are extracted mechanically or chemically from a variety of fruits and seeds.

In the present work, the oil was extracted from the seed of *Citrullus colocynthis* and used to produce a novel PU coating by epoxidation–hydroxylation.

*Citrullus colocynthis* (L.) (CC) is a member of the genus Citrullus within the family Cucurbitaceae, a group often referred to as melons due to their different varieties (Figure 1). The seeds contain approximately 17–19% oil and the oil yield is estimated to be approximately 400 L/ha. CC native from tropical Asia and Africa is now widespread in the Saharo-Arabian phylogeographic region of Africa, in the Mediterranean region, India, and other parts of tropical Asia. This perennial herb is a drought tolerant species, which can survive arid environments by maintaining its water content without any wilting of the leaves or desiccation, even under severe stress conditions and it is able to extend its root system into deep ground water. CCSO has a high percentage of linoleic acid (up to 66%), followed by oleic acid (16.4%), and the total unsaturated fatty acids accounted for over 80% of the seed oil according to the literature [4,5].

Non-edible vegetable oils, especially those with a high unsaturation content (-C=C-), have become more attractive for the further development of polyols owing to their non-food properties, which is essential for safeguarding the global food provision [6]. CCSO is a non-edible oil extracted from the seeds of wild plants that are widespread in KSA. It contains a high content of unsaturated fatty acids, especially linoleic acid (up to 66%), which makes this oil a candidate for use in coating applications.

The unsaturated sites (e.g., -C=C-) and other functional groups in VOs give them the ability to undergo diverse chemical reactions that lead to the production of polymeric materials. Among the various strategies to exploit these double bonds in VOs, one notable approach is the use of the epoxidation reaction, which allows for the incorporation of oxirane oxygen into the double bonds.

Epoxidized vegetable oils serve as essential components in the synthesis of various polymers such as polyesters, epoxy, and polyurethanes, which are frequently utilized as binders in paints. In addition, epoxidized vegetable oils are used as a starting material for the production of a wide range of products, including polyols, which are considered one of the most important compounds in the formation of polyurethanes.

Several studies on the synthesis of epoxidized vegetable oils and biopolyols based on different oils such as soybean [7], linseed [8,9], cottonseed [10], karanja [11], castor oil [12,13], palm oil [14], or rapeseed oil [15], among others [16] have been reported in the literature.

Polyurethanes (PUs), one of the most adaptable and widely used industrial materials, invented by Otto Bayer et al. in 1937, have been effectively used in a variety of applications, including flexible, semi-rigid, and rigid foams [17], elastomers [18], adhesives and composites [19,20], coatings, fibers, thermosets, and thermoplastics [21].

PUs are polymers that contain urethane groups in their structure. They are synthesized by an additional polymerization reaction between isocyanates and polyols (Figure 2). Similar to many polymers, these building blocks of polyurethanes are derived from petrochemicals that are not sustainable. To increase the sustainability of polyurethanes, several studies nowadays have concentrated on substituting petroleum-derived raw materials with renewable resources [21].

The use of oils derived from wild plant seeds represents an untapped renewable resource with potential benefits for both the environment and coatings technology. The coatings industry faces the critical challenge of achieving both environmental sustainability and high-performance functionality. Conventional coatings often rely on petrochemical-derived resins, which raises ecological concerns and limits the scope of sustainable solutions [22,23,24].

In the existing literature, there are no scientific studies on the use of CCSO in the formulation of PU coatings. Therefore, the main aim of this study is to produce biopolyols from CCSO via an epoxidation–hydroxylation route to formulate polyurethane coatings. In addition, this research aims to evaluate the film properties of PU coating using various tests and techniques including the EIS technique to evaluate corrosion performance. The results of this research are expected to lay the foundation for a new era of environmentally conscious protective coatings with improved anti-corrosion properties.

## 2. Materials and Methods

### 2.1. Materials

*Citrullus colocynthis* (CC) fruits were collected from Wadi Hanifa in the Al-Diriyah region (Riyadh, Saudi Arabia) in February 2021. N-hexane (purity ≥ 99% (GC), Honeywell^®^, Seelze, Germany) was used as a solvent. Methanol (purity 99.8% (GC), Sigma Aldrich^®^, Steinheim, Germany), acetone (purity ≥ 99.5%, Sigma Aldrich^®^), aqueous hydrogen peroxide (30 wt.%, Sigma Aldrich^®^), glacial acetic acid (purity ≥ 99.8%, Fisher^®^Chemical, Loughborough, UK), sulphuric acid (95–98%, Panreac^®^, Barcelona, Spain), hydrochloric acid (37%, Panreac^®^), diethyl ether (purity ≥ 99.8%, Sigma Aldrich^®^), and anhydrous magnesium sulfate (purity ≥ 98%, Panreac^®^) were used as supplied. The reagents were required for the determination of hydroxyl and acid value such as potassium hydroxide (purity as KOH ≥ 85%, Loba Chemie^®^, Maharashtra, India), ethanol absolute (purity ≥ 99.97%, CHEM-LAB^®^, Zedelgem, Belgium), n-butanol (purity ≥ 99%, Loba Chemie^®^), pyridine (purity ≥ 99.5%, Loba Chemie^®^), acetic anhydride (purity ≥ 99.7%, VWR^®^, Fontenay-sous-Bois, France), and phenolphthalein indicator (purity ≥ 99%, Loba Chemie^®^). The materials required for film formation, such as BYK-333 as a surface tension reducer, were purchased from Al-Azzaz Chemicals company (Riyadh, Saudi Arabia) and 3HDI isocyanurate from Sigma Aldrich (Sigma Aldrich).

### 2.2. Oil Extraction

To extract oil from the seeds of CC, the seeds were dried in an oven at a temperature of 80 °C for 24 h to reduce their moisture content. Then, the CC seeds were ground using a hammer mill (Karl Kolb, Dreieich, Germany). Subsequently, the oils were extracted from grounded seeds in a Soxhlet apparatus using n-hexane as a solvent. The extraction process was continued at a controlled temperature of 70 °C for 5–6 h. In the second step, the extracted oils were filtered to remove all suspended materials. In the third and final step, hexane was removed for 2 h using a rotary evaporator (Buchii R-215, Flawil, Switzerland) (see Figure 3). The oil content (%) extracted from the CCSO sample was calculated to be 17% according to the following equation:$$\mathrm{Oil}\;\mathrm{content}\left(\%\right)=\frac{W_{\mathrm{oil}}}{W_{\mathrm{solid}}}\;\times \;100$$
where $W_{\mathrm{oil}}$ is the weight of extracted oil and $W_{\mathrm{solid}}$ is the weight of solid sample.

### 2.3. Physicochemical Properties of the Extracted Oils

The density and viscosity of the extracted oils were measured by Anton Paar density meter (DMA 4100M) and viscometer (Lovis 2000 ME).

Iodine and saponification values of oils were obtained from the fatty acid composition of triglycerides by Equations (1) and (2), respectively: [25]
(1)$$\begin{array}{ll} \mathrm{Iodine}\;\mathrm{value}\;\left(\mathrm{triglycerides}\right)= & \left(\%\;\mathrm{Palmitoleic}\;\mathrm{acid}\;\left(C16:1\right)\;\times \;0.950\right)+\\ & \left(\%\;\mathrm{Oleic}\;\mathrm{acid}\;\left(C18:1\right)\;\times \;0.860\right)+\\ & \left(\%\;\mathrm{linoleic}\;\mathrm{acid}\;\left(C18:2\right)\;\times \;1.732\right)+\\ & \left(\%\;\mathrm{linolenic}\;\mathrm{acid}\;\left(C18:3\right)\;\times \;2.616\right)+\\ & \left(\%\;\mathrm{eicosenoic}\;\mathrm{acid}\;\left(C20:1\right)\;\times \;0.785\right)+\\ & \left(\%\;\mathrm{docosenoic}\;\mathrm{acid}\;\left(C22:1\right)\;\times \;0.723\right) \end{array}$$
(2)$$\mathrm{Saponification}\;\mathrm{value}=\frac{\;3\;\times \;56.1\;\times \;1000\;}{(\mathrm{mean}\;\mathrm{molecular}\;\mathrm{weight}\;\times \;3)+92.09\;-\;(3\;\times \;18)}$$
where




Saponification value=mg KOH per g of sample;



3=the number of fatty acids per triacylglycerol;



56.1=MW of KOH (g/mol);



1000=conversion of units (mg/g);

$92.09=\mathrm{MW}\;\mathrm{of}\;\mathrm{glycerol}\;\left(\frac{g}{\mathrm{mol}}\right),\;18=\mathrm{MW}\;\mathrm{of}\;\mathrm{water}\;(g/\mathrm{mol}$).


The acid value of the oils was determined quantitatively by titration with 0.1 M KOH in methanol solution and phenolphthalein indicator. About 5 g of oil was weighed and dissolved in 50 mL of absolute ethanol, then 2–3 drops of phenolphthalein indicator solution were added. The mixture was then heated in a water bath for 10 min with constant stirring. The titration was stopped when the color changed to a faint pink. The acid value was then calculated using the following equation:$$\mathrm{Acid}\;\mathrm{value}=\;\frac{V\;\times \;N\;\times \;56.1}{\mathrm{Weight}\;\mathrm{of}\;\mathrm{sample}}$$
where V is the volume of KOH used for titration, mL, N is the normality of KOH, and 56.1 denotes the potassium hydroxide molecular weight (g/mol).

### 2.4. Epoxidation Process

The epoxidation reaction was carried out in a three-neck flask equipped with a magnetic stirrer (IKA C-MAG HS-7, IKA, Staufen, Germany).

The experimental procedures of this process were carried out as follows:

At the beginning, 30 g of each oil was added to the flask with a calculated amount of glacial acetic acid. The mixture was kept at a constant temperature (60 °C) and a suitable stirring speed that allowed for sufficient agitation to ensure proper mass transfer. After 10 min, sulfuric acid and hydrogen peroxide were mixed and added dropwise to the mixture. The addition was completed within 30 min at a constant rate according to the method described by Dinda et al. [16,26]. The reaction was allowed to continue for 6 h after addition of the acids (see Figure 4).

The final product was then washed with water and extracted with petroleum ether. Then, petroleum ether was removed in a rotary evaporator [16,27]. The conversion (%), epoxidation (%), and selectivity (%) of the produced epoxidized oil were calculated as described in the literature [28,29].

### 2.5. Hydroxylation Process

The CCSO-polyol was synthesized according to the method described by Hazmi et al. [30] with some modifications. The molar ratio of epoxy groups to methanol to distilled water was 1:11:2. The concentration of the catalyst was 1 wt.% of the total weight of distilled water and methanol. The calculated weights of methanol, distilled water, and sulfuric acid were added to a 250 mL three-neck flask equipped with a magnetic stirrer, a thermocouple, and a reflux condenser. The mixture was stirred at a constant speed at 65 °C. Then, 15 g of the prepared epoxidized oil was added to the reaction flask. The reaction was then continued for 2 h. The resulting mixture was then washed with distilled water and extracted with petroleum ether in a 250 mL separatory funnel to remove the unreacted catalyst, followed by the removal of the excess water, methanol, and petroleum ether using the rotary evaporator. The resulting polyol was measured for its hydroxyl value (ASTM E222 [31]), acid value (ASTM D7253-16 [32]), and analyzed by FT-IR and ^1^H NMR [33].

### 2.6. PU Film Formulation and Casting

The CCSO-based PU films were formulated with a molar ratio of NCO to OH of 1:1. First, a calculated amount of the prepared polyol was added to a screw bottle (50 mL) and diluted with an appropriate amount of methyl ethyl ketone (MEK). Drops of BYK-333 were then added to the mixture as a surface tension reducer. The mixture was then stirred for 5 min to ensure a homogeneous mixture. Prior to coating, the surface of the glass panels (12 cm × 12 cm) was cleaned with acetone. The homogeneous mixture was uniformly coated over glass panels using an eight-sided square film applicator (80 mm wide, gap sizes 1–8 mils, high technology mart Instruments, New York, NY, USA) with 105 µm gap size (4 mils) and kept at room temperature for 1 h to allow the solvent to evaporate before curing at 110 °C for 2 h. The same procedures were followed for casting over aluminum panels (Al 1001, 12.5 cm × 25 cm). To investigate the effect of surface pretreatment on the film properties, the surface of the aluminum substrate was polished with sandpaper (1200 grit) before casting the 4 mm thick wet film based on CCSO. The hardness of the films was tested at various time intervals. Once a suitable hardness was achieved, the films were used for further analysis. The formulations of the PU coatings were summarized in Table 1.

### 2.7. Characterization

#### 2.7.1. Fatty Acid Composition

Fatty acid analysis was performed using a GC–MS chromatograph (QP2010 Ultra, Shimadzu, Tokyo, Japan). Fatty acid methyl esters were prepared according to the method described by Nehdi et al. [5].

About 0.1 g of oil was converted to methyl esters using 0.5 mL of CH_3_ONa (2M) and 0.5 mL of HCl (2M) in 2.5 mL of hexane. The GC was equipped with an FID detector and an Rxi-5Sil MS column (30 m × 0.25 mm inner diameter, 0.25 µm film thickness). The temperatures of the detector and the injector were programmed at 275 and 220 °C, respectively. The temperature gradient of the column ranged from 150 to 180 °C at 15 °C/min, followed by an increase to 210 °C at 1 °C/min. Helium was used as the carrier gas (1.5 mL/min). The MS was operated in electron ionization (EI) mode at 70 eV. The peaks were identified by matching their mass spectra with the peaks in the Wiley library database [4].

#### 2.7.2. Spectroscopy Analysis

The FT-IR spectra of the CCSO, epoxidized CCSO (ECCSO), synthesized CCSO-polyol, and CCSO-PU coating were recorded on a SHIMADZU FOURIER transform infrared spectrometer using KBr pellet at wavelengths between 4000 and 400 cm^−1^ and at a scan rate of 4 cm^−1^.

The ^1^H NMR spectra of the CCSO and ECCSO were recorded in deuterated chloroform (CDCl_3_) at the resonance frequency of 400 MHz using a JEOL ECLIPSE 400 spectrometer (Tokyo, Japan).

#### 2.7.3. Thermal Analysis

Thermal analyses were carried out to investigate the thermal degradation of the CCSO-PU coating. In the experiment, approximately 5–10 mg of the sample was heated in a nitrogen atmosphere at a heating rate of 10 °C/minute in a temperature range from 25 °C to 600 °C using a STA 449 F1 Jupiter^®^ (Netzsch, Selb, Germany) thermogravimetric analyzer (TGA). The results of the analysis were plotted in the form of weight loss as a function of temperature. To determine the glass transition temperature (Tg) of the CCSO-PU coating, a specific methodology was employed utilizing the differential scanning calorimetry (DSC TA Instruments 60 WS, Tokyo, Japan). The sample was subsequently cooled to −50 °C and allowed to remain at this temperature for 3 min before being heated to 150 °C at a rate of 10 °C/min in a nitrogen atmosphere. During this process, the variations in heat flow in response to the temperature changes were accurately recorded.

#### 2.7.4. Coating Tests

The coating properties were subjected to a comprehensive evaluation through standard tests, including chemical resistance, gloss, pencil hardness, adhesion, pendulum hardness, and EIS tests.

Chemical resistance was determined according to ASTM D 543-95 [34] by immersing approximately (10–11) g of the PU in separate containers, containing a 10% HCl solution and a 1% NaOH solution for a period of seven days. In addition, the gloss of the resulting PU coating was evaluated according to ASTM D523-14 [35] using a trigloss meter (Sheen Instruments, Model 260, Boston, MA, USA). This carefully calibrated instrument was placed on the sample surface and operated in an automated manner to calculate gloss measurements at reflection angles of 20°, 60°, and 85° [36].

The evaluation of the pencil hardness of the produced polyurethane (PU) film was performed using the pencil hardness tester (Sheen Instruments Model SH720N) at room temperature according to ASTM D3363-05 [37]. A series of pencils with different degrees of hardness, ranging from soft (9B) to hard (9H), were used for the evaluation. The pencil hardness of the test sample was considered to be the pencil hardness that did not result in any form of damage to the coating, such as scratching or cracking.

The adhesion test for the produced PU coating was determined based on the guidelines provided by the ASTM 3359 [38], which requires the measurement of coating adhesion through a tape test. In this assessment, cross-shaped incisions were made on the coating using a tool equipped with sharp blades, forming a pattern resembling a ‘+’. Subsequently, pressure-sensitive tape was applied and removed. The incisions were then assessed in accordance with a standard table.

The Pendulum test, as prescribed by the ASTM D4366 [39], was employed to determine the damping time or number of oscillations exhibited by a pendulum that oscillates on a test surface. In this test, the pendulum is positioned on the surface of the coating and adjusted to oscillate between a deflection angle ranging from 12° to 4°. Throughout this period, the numbers of pendulum oscillations are accurately counted and relayed through an electronic device. The higher the count of oscillations, the greater the level of hardness.

#### 2.7.5. Electrochemical Impedance Spectroscopy (EIS) Test for Corrosion

Electrochemical impedance spectroscopy (EIS) is an extremely valuable technique for evaluating and monitoring the degradation or expansion rate of polymeric coatings when exposed to electrolytes. In addition, it facilitates the exploration of the electrochemical cathodic and anodic reactions that take place during the corrosion process. EIS offers numerous remarkable advantages in the field of polymer coating analysis. First and foremost, it is non-destructive, enabling the assessment of coating degradation or regeneration. Most importantly, this technique provides quantitative data on the electrochemical processes in the system [40].

In this study, an EIS frequency sweep was performed at a potential that was 10 mV greater than the open circuit potential and covered a range from 100 kHz to 100 mHz. A cylindrical glass cell with a diameter of 4.9 cm was mounted on an aluminum panel (Al 1001, 12.5 × 25 cm). This panel was covered with a wet PU coating of 4 mil (105 µm). The electrochemical cell was filled with 60 mL of a 3.5 wt.% NaCl electrolyte. The reference electrode was a silver/silver chloride electrode, the counter electrode was platinum, and the working electrode was the panel coated with the PU material.

## 3. Results and Discussion

### 3.1. Fatty Acid Compositions

The fatty acid compositions of CCSO were determined using a GC–MS instrument and the results are summarized in Table 2. The fatty acid profile of CCSO revealed that the major fatty acid is linoleic acid (C18:2; 68.38%), followed by oleic acid (C18:1Δ9; 14.22%), palmitic acid (C16:0; 9.41%), stearic acid (C18:0; 6.86%), and traces of oleic acid (C18:1Δ11) and linolenic acid (C18:3). The total proportion of unsaturated fatty acids in CCSO was 83.21%.

### 3.2. Physicochemical Properties of the Extracted Oils

The physicochemical properties of CCSO are shown in Table 3. CCSO was a pale-yellow liquid with a molecular weight of 873.68 g/mol and an iodine value of 131.61 g. The high iodine value indicates that CCSO has many double bonds in its chemical structure, making it a very interesting chemical feedstock for a variety of modifications. The acidity and saponification values of CCSO were 0.97 mg KOH/g and 193.45 mg KOH/g, respectively.

### 3.3. Spectroscopy Analysis

#### 3.3.1. Fourier-Transform Infrared Spectroscopy (FT-IR)

Evidence of the formation of epoxidized oil (EO) and biopolyols was provided by FT-IR spectroscopy. Comparison of the FT-IR spectra (Figure 5) of the original oil (CCSO), the epoxidized oil (ECCSO), and the corresponding bio-polyol (CCSO-polyol) showed the presence of bonds characteristic of the structure of fatty acid glycerides. The FT-IR spectra of all epoxidized oils and the corresponding polyols were almost identical to those of the original oil with only minor differences (see Figure 5). In particular, the spectrum of the epoxidized oil exhibited doublet bands at 844 cm^−1^ and 823 cm^−1^ due to the presence of epoxy groups [16,41], which were not present in the spectra of CCSO and its biopolyol. In addition, a low intensity band at 3454 cm^−1^ was observed in the spectrum of the epoxidized oil, indicating the stretching vibrations of O–H of hydroxyl groups. However, the low intensity indicates the presence of side reactions during the synthesis of the epoxidized oil. On the other hand, the FT-IR spectrum of CCSO-polyol showed a high intensity of the band at 3454 cm^−1^, indicating a significant number of O–H bonds. Furthermore, the band of the C–H bond of the olefin groups (3008.95 cm^−1^) and the pendulum vibrations of the CH2 groups (725 cm^−1^) were considerably disappeared or reduced in the oil spectrum. The stretching vibration of the C=C bond of the olefin group, which was observed in the oil spectrum at 1650 cm^−1^, has also disappeared [41]. The most important absorption bands of CCSO, epoxidized CCSO, and its biopolyols are shown in Figure 5.

The Fourier transform infrared spectrum of polyurethane coatings based on CCSO is shown in Figure 6. The absence of the band in the range from 2260 to 2310 cm^−1^ confirms the absence of a free NCO group in the polymer structure, and thus the completion of the urethane reaction. The main recognizable features of the polyurethane are the bands at 1095.4 cm^−1^ (C–N stretching vibrations), 1126 cm^−1^ (C–O stretching vibrations), 1373.13 and 1519.69 cm^−1^ (C–N stretching and N–H bending), 1681.69 cm^−1^ (C=O stretching vibrations originating from urethane groups), 1735.69 cm^−1^ (C=O stretching vibrations of fatty acid), 2854.3–2923.7 cm^−1^ (symmetric and asymmetric CH_2_ stretching vibrations), and 3386 cm^−1^ (N–H stretching vibrations of urethane groups). The presence of these vibrational bands substantiates the formation of the urethane linkage NH–COO in the synthesized polyurethane [42,43,44].

#### 3.3.2. ^1^H NMR Spectra

To confirm the changes in functional groups in ECCSO compared to the original oils, ^1^H NMR spectra were generated. In Figure 7, the ^1^H NMR spectra of CCSO and its epoxidized oil (ECCSO) were plotted and show a signal intensity (A) at 5.2–5.5 ppm. This characteristic peak corresponds to the vinyl hydrogens of the double bonds that have almost disappeared in the sample of epoxidized oil (ECCSO) due to the conversion of the double bonds into epoxy groups. This peak was split into two signals: a small signal for the vinyl hydrogens of the double bonds at 5.4 ppm, indicating that only a few double bonds remained after the epoxidation reaction, and a signal for the central hydrogen of the glyceride moiety at 5.25 ppm [45]. The peaks at 4.1–4.3 ppm (B) represent the four hydrogens of the methylene groups of the glycerol moiety. In addition, two other signals were observed at 2.0 ppm (D), associated with hydrogen near double bonds, and at 2.8 ppm (H), representing allylic hydrogen between double bonds (Figure 7 CCSO). The peaks at 2.3 ppm (C) associated with the methylene hydrogens adjusted to the carbonyl group, 1.6 ppm (E) and 1.3 ppm (F) are the peak areas associated with the methylene hydrogens of the triglyceride, and 0.95 ppm (G) is the peak area of the methyl hydrogens of the triglyceride. In the spectrum of the epoxidized oil, I and J are the peak areas associated with the hydrogens of the epoxide groups and occur at chemical shifts of 2.9 (monoepoxide) and 3.1 ppm (diepoxide), respectively. The detection of hydroxyl groups resulting from the opening of the oxirane ring was observed in the range of 4–3.4 ppm, as described by M. Farias et al. [29]. The ^1^H NMR spectra obtained were consistent with the previous analysis, confirming the formation of oxirane oxygen in the double bonds of the ECCSO sample as well as hydroxyl groups as a result of side reactions.

### 3.4. TGA and DSC Analysis

The thermal stability of PU coatings based on CCSO was investigated using a TGA equipment. Figure 8 shows the TGA thermogram of the CCSO-PU coating over the temperature range from 25 °C to 600 °C. The thermal degradation of the CCSO-PU coating showed only one stage of weight loss. Approximately 5% of the weight of each specimen was lost at the temperature of 322 °C. The major weight loss of the tested sample occurred at a temperature higher than 485 °C. Table 4 summarizes the thermal analysis of the PU coatings at different weight losses.

The glass transition temperature (Tg) values of CCSO-PU coating were recorded to be 10.2 °C (see Table 5).

### 3.5. Coating Tests

#### 3.5.1. Chemical Resistance

The CCSO-based PU coating was immersed in separate containers containing 10% HCl and 1% NaOH solutions for seven days, to assess its chemical resistance. As shown in Table 5, the results showed that the CCSO-PU coating exhibited excellent resistance to both acidic and alkaline conditions.

#### 3.5.2. Physical Properties of Alkyd Coatings

The physical properties of the PU coatings based on CCSO are listed in Table 6. The pencil hardness was tested from 9B to 9H, with 9H having the highest hardness. Based on the hardness value of the test pencil, the pencil hardness value of the CCSO-PU coatings was rated as (HB).

Gloss measurement is an important property of a coating that results from the interaction between light and the surface of the coating film [4]. The gloss of the CCSO-PU coatings at 20°, 60°, and 85° was found to be excellent. As summarized in Table 6, the pendulum hardness of CCSO-PU was measured to be 55 oscillations.

Table 6 shows that the CCSO-PU coatings had a cross-hatch adhesion 0B grade prior to surface treatment. However, pretreatment of the CCSO-PU substrate surface resulted in a significant improvement in cross-hatch adhesion, which reached to 5B grade. In addition, the hardness of the CCSO-PU coating improved and reached a level of 4H.

#### 3.5.3. EIS Analysis

Figure 9 shows the spectra of electrochemical impedance spectroscopy (EIS) of CCSO-PU coatings after surface pretreatment (CCSO1200-PU) in Bode diagram format. In the Bode diagram, the modulus, which represents the absolute value of the impedance, is plotted on a logarithmic scale along the y-axis. The logarithm of the frequency from 100,000 to 0.1 Hz is shown on the x-axis.

The CCSO-based PU coating was applied to an aluminum substrate after the surface had been mechanically pre-treated with 1200 grit sandpaper. The coating showed excellent capacitive behavior with an initial impedance of 1.08 × 109 Ω at 0.1 Hz. The resistance of the coating gradually decreased to 7.99 × 108 Ω after being exposed for 23 days. A single time constant was observed for the CCSO1200 PU coating, and the coating systems can be represented by a Randles cell equivalent circuit, as shown in Figure 10a.

## 4. Conclusions

A novel bio-polyol was obtained from CCSO in a two-step process. In the first step, the unsaturated fatty acid was epoxidized in an in situ generated peracetic acid. In the second step, the resulting epoxy groups were opened with methanol. The chemical structure of the synthesized polyol was confirmed by FT-IR and ^1^H NMR spectroscopy. This bio-polyol based on CCSO showed a hydroxyl value of 201.1 mg KOH/g and an acid value of 29.31 mg KOH/g. It was then used to formulate PU coatings using 3HDI isocyanurate as a crosslinker.

The resulting CCSO-PU coatings were subjected to comprehensive characterization, including FT-IR spectroscopy, TGA, DSC, chemical resistance, pencil hardness, gloss, pendulum hardness, and cross-hatch adhesion. In addition, their corrosion performance was evaluated by EIS. The CCSO-PU coatings showed exceptional chemical resistance in both acidic and alkaline environments. They exhibited moderate pencil hardness (HB) and excellent gloss at an angle of 60°. EIS analysis showed high resistivity and stability at low frequencies after exposure to a 3.5% NaCl electrolyte for 23 days. After mechanical surface pretreatment of the CCSO-PU substrate, the film properties improved significantly, with pencil hardness increasing to 4H and cross-hatch adhesion to 5B. In summary, the use of CCSO for bio-polyol synthesis presents significant potential. This approach is consistent with the principles of sustainable development and green chemistry as it enables the substitution of petrochemical polyols. The results show that CCSO is a promising innovative bio-based feedstock for coating applications.

## Figures and Tables

**Figure 1 polymers-16-00214-f001:**
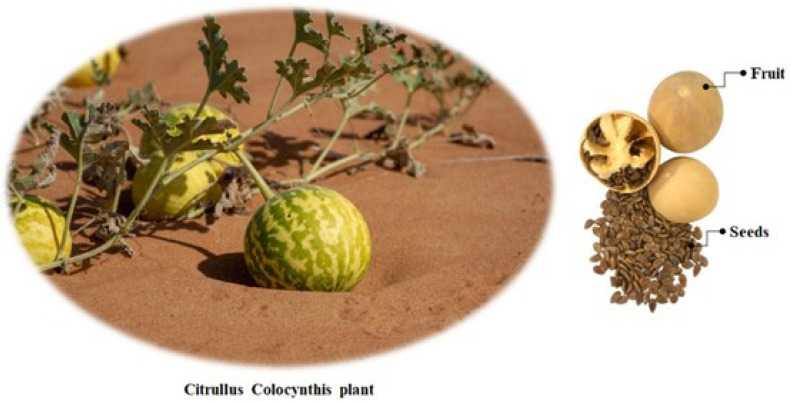
*Citrullus colocynthis* (L.) plant and its seeds.

**Figure 2 polymers-16-00214-f002:**
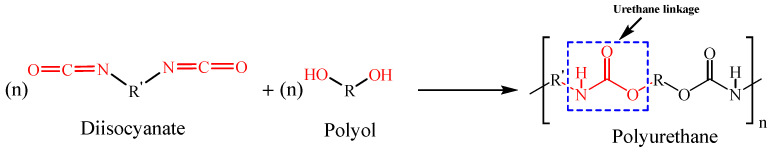
Preparation of PU from polyol and diisocyanate.

**Figure 3 polymers-16-00214-f003:**
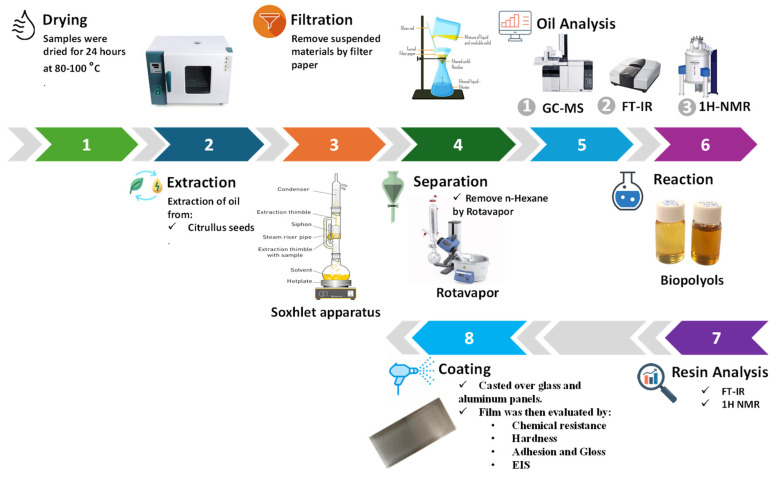
Graphic illustration of the experimental procedures.

**Figure 4 polymers-16-00214-f004:**
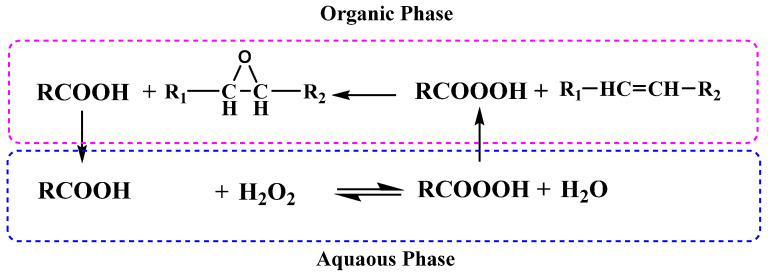
Epoxidation process of VOs with acetic acid.

**Figure 5 polymers-16-00214-f005:**
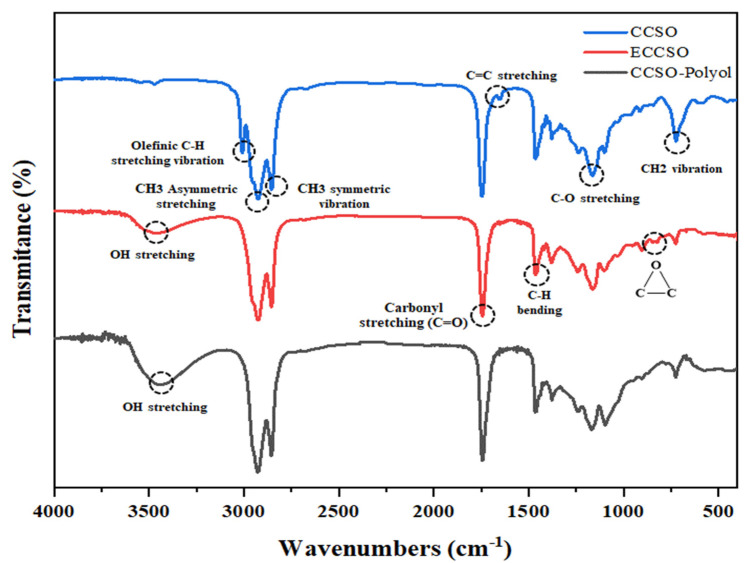
FT-IR spectrum of CCSO, ECCSO, and CCSO-polyol.

**Figure 6 polymers-16-00214-f006:**
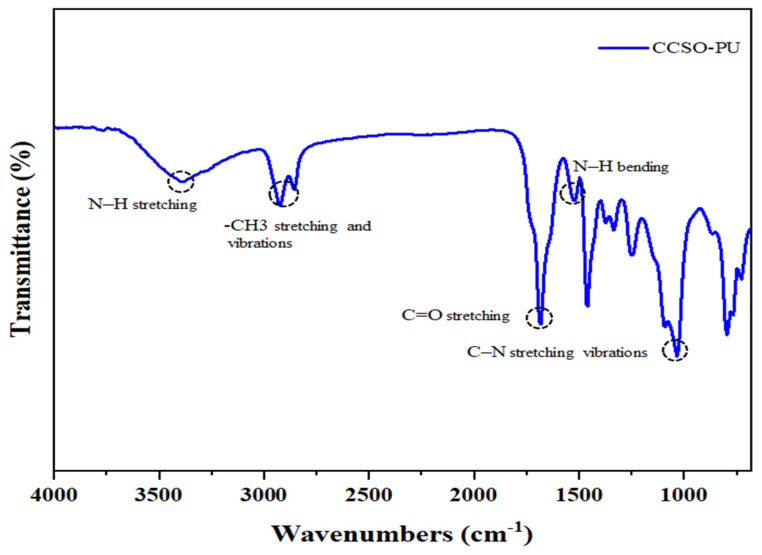
FT-IR spectrum of PU coating based on CCSO.

**Figure 7 polymers-16-00214-f007:**
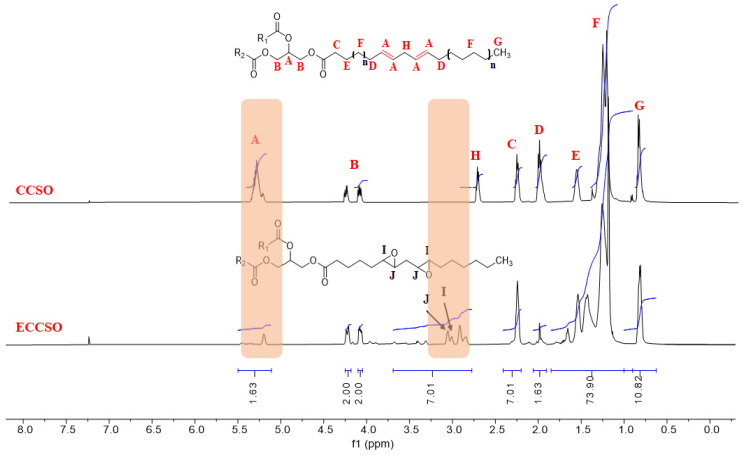
^1^H NMR spectra of CCSO and ECCSO.

**Figure 8 polymers-16-00214-f008:**
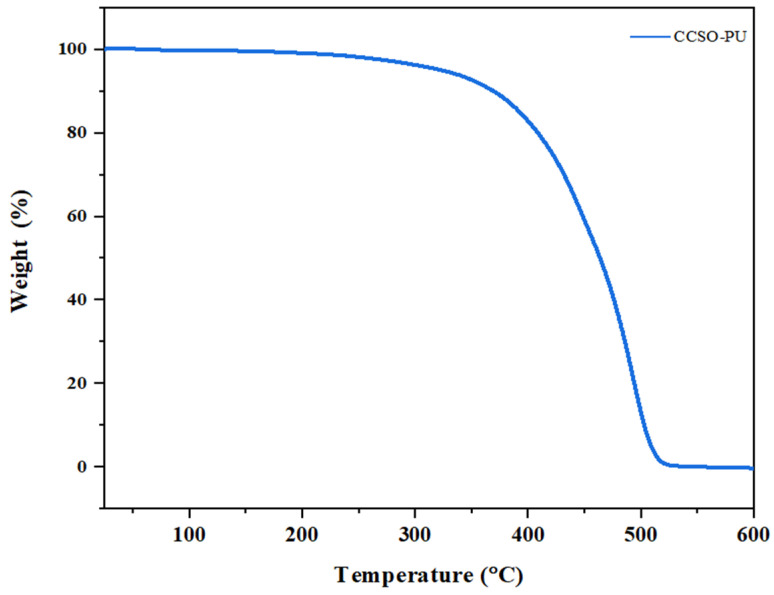
TGA thermogram of PU coating based on CCSO.

**Figure 9 polymers-16-00214-f009:**
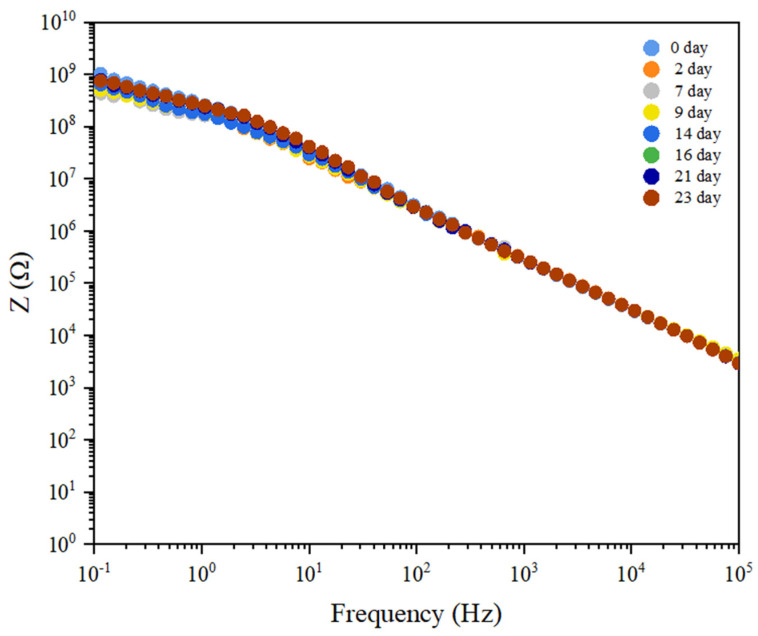
Bode plot for CCSO1200-PU coating as a function of immersion time.

**Figure 10 polymers-16-00214-f010:**
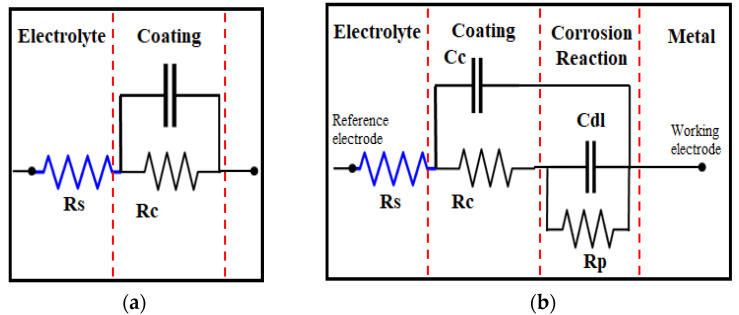
Randles cell equivalent circuit (**a**) and equivalent circuit for corrosion (**b**).

**Table 1 polymers-16-00214-t001:** Formulation of PU coatings based on CCSO.

Coating	CCSO-PU
Polyol (wt.%)	47
3HDI (wt.%)	33
Solvent (wt.%)	18
BYK-333 (wt.%)	2

**Table 2 polymers-16-00214-t002:** Fatty acid compositions of CCSO.

Fatty Acids	Compositions (%)
	CCSO
**Saturated**	16.27
Palmitic (C16:0)	9.41
Stearic (C18:0)	6.86
Total unsaturated	83.21
**Monounsaturated**	14.59
Oleic (C18:1Δ9)	14.22
Oleic (C18:1Δ11)	0.37
**Polyunsaturated**	68.62
Linoleic (C18:2)	68.38
Linolenic (C18:3)	0.24

**Table 3 polymers-16-00214-t003:** Physiochemical properties of CCSO.

Properties	CCSO
Color	Yellow
Oil yield (*w*/*w*) (%)	17
Acid value (mg KOH/g)	0.97
Iodine value (I_2_ g/100 g oil)	131.61
Saponification value (mg KOH/g)	193.45
Density (25 °C) (g/cm^3^)	0.9035
Kinematic viscosity (mm^2^/s)	-
Dyn. Viscosity mPa.s	31.65
Molecular weight (g/mol)	873.68

**Table 4 polymers-16-00214-t004:** Thermal analysis of produced PU coatings at different weight losses (%).

Coatings	CCSO-PU
Td5 (°C)	322.7
Td15 (°C)	392.3
Td30 (°C)	431.7
Td50 (°C)	463.7

**Table 5 polymers-16-00214-t005:** Glass transition temperatures and chemical resistance results of produced PU coatings.

Type of Coating	Tg (°C)	Chemical Resistance	
		Acidic (10% HCl)	Basic (1% NaOH)
CCSO-PU	10.2	Excellent	Excellent

**Table 6 polymers-16-00214-t006:** Physical properties of PU coatings based on CCSO.

Coating	Persoz Hardness (osc.)	Gloss			Pencil Hardness		Cross-Hatch Adhesion	
		20°	60°	85°	Before	After	Before	After
CCSO-PU	55	129	172	96.3	HB	4H	0B	5B

## Data Availability

Data are contained within the article.

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
