# Peer review of "Preparation of Bio-Based Polyurethane Coating from Citrullus colocynthis Seed Oil: Characterization and Corrosion Performance"

_polymers, 2024, doi:10.3390/polym16020214_

Round 1

Reviewer 1 Report

Comments and Suggestions for Authors

1.     Keywords are not relevant or unnecessary. See eg only one out of epoxidation and epoxy is appropriate. Similarly, it should be ‘surface coating’ not ‘surface coatings’

2.     What makes VO of Citrullus colocynthis a candidate of choice for use as foam coating? It needs to be elaborated.

          3.     Lots of work has been reported on VO-based coatings including reviews. However, the authors did not mention many relevant citations in the Introduction section. See for example https://doi.org/10.1016/j.porgcoat.2021.106267

4.     Thus, the novelty, objectives, and significance of the reported work needs to be elaborated.  

5.     The major concern of the reviewer is that the authors have not discussed well the positives of this work and how it is an advance over the previously reported ones. So, they need to improve the discussion of results and compare with the studies reported in the literature.  

6.     The preceding statement applies to characterization studies also.

7.     The Para ‘The utilization of oils …. solutions.’ Requires to be enriched with citations.

8.     Language needs authors good attention. See ‘Citrullus Colocynthis’ instead of Citrullus colocynthis., etc.

Comments on the Quality of English Language

Average. Needs improvement.

Reviewer 2 Report

Comments and Suggestions for Authors

The authors have worked on the preparation of Bio-based Polyurethane coating from Citrullus Colocynthis seed oil in regard to the characterization and corrosion performance. 

1. Introduction: Some remarks like 'epoxides are synthesized from different seed oils like linseed, soybean etc and are used in different industrial polymer products like composites, coatings, tires etc. However, what sets apart this research is the use of Citrullus Colocynthis Seed Oil epoxides. 

Also, more reviews referring to and citing seed oil epoxides would be beneficial. Example: https://doi.org/10.1021/acs.iecr.2c03867

2.  Further, we would be able to find and cite few literature that talks about seed oil based polyurethanes, mainly castor oil based. https://doi.org/10.1016/j.reactfunctpolym.2022.105496

3. Peracid route for epoxidation is the most common, however, had the authors tried other epoxidation methods like AIER, metal catalyst, enzymes etc? If you can state why peracid route was chosen, that would be great. 

4. Could you please specify the epoxy equivalent weight? (EEW) Did I miss it? EEW needs to be in the manuscript if someone would like to repeat or continue this research. 

Round 2

Reviewer 1 Report

Comments and Suggestions for Authors

The reviewer's commnets on the original manuscript have been addressed.

Comments on the Quality of English Language

Some minor checks are needed.